# Repurposing Anti-Dengue Compounds against Monkeypox Virus Targeting Core Cysteine Protease

**DOI:** 10.3390/biomedicines11072025

**Published:** 2023-07-18

**Authors:** Mohd Imran, Nawaf M. Alotaibi, Hamdy Khamees Thabet, Jamal Alhameedi Alruwaili, Lina Eltaib, Ahmed Alshehri, Ahad Amer Alsaiari, Mehnaz Kamal, Abdulmajeed Mohammed Abdullah Alshammari

**Affiliations:** 1Department of Pharmaceutical Chemistry, College of Pharmacy, Northern Border University, Rafha 91911, Saudi Arabia; abeda.mohammed@nbu.edu.sa; 2Department of Clinical Pharmacy, College of Pharmacy, Northern Border University, Rafha 91911, Saudi Arabia; 3Chemistry Department, College of Arts and Sciences, Northern Border University, Rafha 91911, Saudi Arabia; 4Medical Lab Technology Department, College of Applied Medical Sciences, Northern Border University, Arar 91431, Saudi Arabia; 5Department of Pharmaceutics, College of Pharmacy, Northern Border University, Rafha 91911, Saudi Arabia; 6Department of Pharmacology and Toxicology, College of Pharmacy, Northern Border University, Rafha 91911, Saudi Arabia; 7Department of Pharmacology, College of Clinical Pharmacy, Imam Abdulrahman Bin Faisal University, King Faisal Road, Dammam 31441, Saudi Arabia; 8Department of Clinical Laboratory Sciences, College of Applied Medical Sciences, Taif University, P.O. Box 11099, Taif 21944, Saudi Arabia; 9Department of Pharmaceutical Chemistry, College of Pharmacy, Prince Sattam Bin Abdulaziz University, Al-Kharj 11942, Saudi Arabia; m.uddin@psau.edu.sa; 10College of Pharmacy, Northern Border University, Rafha 91911, Saudi Arabia

**Keywords:** monkeypox virus, cysteine proteinase, in silico methods, molecular docking, molecular dynamic simulations

## Abstract

The monkeypox virus (MPXV) is an enveloped, double-stranded DNA virus belonging to the genus Orthopox viruses. In recent years, the virus has spread to countries where it was previously unknown, turning it into a worldwide emergency for public health. This study employs a structural-based drug design approach to identify potential inhibitors for the core cysteine proteinase of MPXV. During the simulations, the study identified two potential inhibitors, compound CHEMBL32926 and compound CHEMBL4861364, demonstrating strong binding affinities and drug-like properties. Their docking scores with the target protein were −10.7 and −10.9 kcal/mol, respectively. This study used ensemble-based protein–ligand docking to account for the binding site conformation variability. By examining how the identified inhibitors interact with the protein, this research sheds light on the workings of the inhibitors’ mechanisms of action. Molecular dynamic simulations of protein–ligand complexes showed fluctuations from the initial docked pose, but they confirmed their binding throughout the simulation. The MMGBSA binding free energy calculations for CHEMBL32926 showed a binding free energy range of (−9.25 to −9.65) kcal/mol, while CHEMBL4861364 exhibited a range of (−41.66 to −31.47) kcal/mol. Later, analogues were searched for these compounds with 70% similarity criteria, and their IC_50_ was predicted using pre-trained machine learning models. This resulted in identifying two similar compounds for each hit with comparable binding affinity for cysteine proteinase. This study’s structure-based drug design approach provides a promising strategy for identifying new drugs for treating MPXV infections.

## 1. Introduction

The appearance of a new epidemic caused by the monkeypox virus (MPXV) has urged public health professionals to be concerned about its potential to pose a new threat following the COVID-19 pandemic [1]. The monkeypox virus (MPXV) is a species that belongs to the genus Orthopox viruses, which includes the variola, cowpox, and vaccinia viruses [2,3,4,5]. MPXV has a double-stranded DNA molecule and was identified in monkeys [6]. On the other hand, rope squirrels, tree squirrels, and Gambian rats are all natural hosts for the MPXV virus [7]. There have been findings of two clades of the monkeypox virus in both central and west Africa, with the first clade causing a more severe form of the illness [8]. As per a study conducted by Bunge et al., the mortality rates associated with different monkeypox strains observed in Africa range from 3.6% to 10.6% [9]. Transmission of the MPXV may occur via physical contact with contagious lesions, scabs, bodily secretions, or shared clothing.

Moreover, many infections have been attributed to sexual contact in numerous outbreaks [8]. The cases in a few outbreaks have been atypical, with a distinctive rash beginning in the vaginal and perineal regions and spreading to other body parts [10]. Patients are considered infectious when the rash appears on their bodies and remains so until the lesions scab over and eventually fall off. The symptoms are comparable to those of smallpox, but the severity of the disease is less severe. The rash is preceded by less severe symptoms such as fever, lymphadenopathy, and symptoms that are similar to the flu [11]. To complete their reproductive cycles, most viruses, such as the human immunodeficiency virus, adenovirus, and poliovirus, require post-translational proteolytic processing. For the orthopoxvirus to be able to produce infectious progeny, the orthopoxvirus core proteins must first undergo the process of proteolytic maturation [12].

In contrast to other DNA viruses, poxviruses typically replicate in the cytoplasm of infected cells by utilizing various virus-encoded proteins, such as cysteine proteinase [13]. The proteases cleave the precursor polyproteins, which are essential in the viral replication cycle. As it was considered that these enzymes are crucial for the replication of viruses, it is possible to develop a treatment that involves the inhibition of viral protease by using active chemical substances [14,15]. The cysteine (I7L) core proteinase cleaves viruses’ primary structural and membrane proteins [16]. Proteases like these are recognized as some of the most promising therapeutic targets. In other viruses, including HIV, protease inhibitors have demonstrated promising viral protease inhibition due to their important role in viral replication, by cleaving pioneer polyproteins [17]. The core proteinase (I7L) activity can be inhibited by a compound known as TTP-6171 [18,19]. However, mutations in viral proteins might also create drug-resistant strains [16]. A recent study focused on the cysteine proteinase of MPXV, and the antiviral drugs fosdagrocorat and lixivaptan were investigated [18]. Moreover, computational repurposing drug design studies for tetracycline to target cysteine proteinase were demonstrated in another study [20]. Similar in silico studies have also been performed on coronaviruses [21]. Another study demonstrated that cysteine proteinase could be a potential target for the discovery of a new drug against the MPXV [22].

The development of drugs applying a structure-based approach is currently challenging due to the unavailability of the experimental structure of cysteine proteinase in MPXV. In addition to the structure-based method, computational chemistry and chemoinformatics can also be utilized to develop inhibitors for cysteine proteases. These complementing techniques include, among others, experimental design for chemical libraries and screening sets, methods for analyzing HTS and other screening data, and QSAR methods [23]. However, this study employed computational techniques to generate an in silico model of the viral core cysteine proteinase of MPXV. This study applied an ensemble docking approach to investigate anti-dengue compounds against MPXV targeting core cysteine protease (I7L), by screening the anti-dengue compound library. Initially, the cysteine proteinase of MPXV was modelled and simulated for 100 ns to refine the protein structure. Later, clustering was performed, and the most plausible structures were identified. Ensemble docking was performed by targeting different conformations of the binding site. Further, molecular dynamics simulation (MD) was applied to examine the binding pattern and stability of the best hits for each binding pocket. This examination identified a group of promising hit compounds that can be deployed as a therapeutic option against the MPXV virus and could be validated in the in vitro experiment.

## 2. Methodology

### 2.1. Protein Structure Modelling

The experimental determination of the 3D structure of the MPXV cysteine proteinase remains unresolved; therefore, a computational model of the protein structure was constructed in this study. The FASTA sequence of the MPXV cysteine proteinase was sourced from the NCBI protein database under the GenBank ID: UXF48136.1 [24]. Subsequently, AlphaFold Colab v2.1.0, a computational tool using machine learning and multiple sequence alignment (MSA) techniques, was employed to generate a 3D structure model of the MPXV cysteine proteinase [25]. The FASTA sequence served as the input to AlphaFold Colab v2.1.0, producing the 3-dimensional structure of MPXV cysteine proteinase. The AlphaFold and ProTSAV 2004–2020 (Protein Structure Analysis and Validation) provided the predicted structure’s reliability in terms of confidence score [26].

### 2.2. Molecular Dynamics Simulation of Modelled Structure

Further, the AlphaFold-generated protein structure of MPXV cysteine proteinase was deployed in a molecular dynamics (MD) simulation, followed by clustering analysis to determine the most probable conformations. The GROMACS 2021.7 software package performed the molecular dynamics simulation [27,28,29]. The CHARMM36 force field was used to generate topology parameters (charge, radius, and epsilon) that were subsequently assigned to all the protein atoms [30]. The long-range electrostatic interactions were computed using the Ewald Particle Mesh method [31]. The system was further neutralized by introducing Na+ and Cl- ions and solvated with the TIP3P water cube model. Subsequently, the system was made to undergo 50,000 steepest descent minimization steps and then heated to 310 K for 1 ns to achieve the NVT equilibrium ensemble state. At that point, the system was equilibrated at a constant pressure of 1 bar in the NPT ensemble state. The LINCS (Linear Constraint Solver) algorithm was utilized for the purpose of putting constraints on the covalent and hydrogen bonds of the protein [32]. Post equilibrium, a 100 ns production run was performed to enable the system to undergo conformational changes. The GROMACS g_cluster package was employed for clustering within the RMS cutoff of 0.3 nm used with the Gromos cluster method. The representative and central structure of the clusters with the most members was selected to perform ensemble docking. Here, the criteria used to select the cluster were that the number of structures should be greater than 10% of the total conformations generated.

### 2.3. Compound Library

The anti-dengue compounds were obtained from the ChEMBL database, which is a publicly accessible repository maintained by the European Bioinformatics Institute (EMBL-EBI) [33]. The database was searched for drug targets that have been reported to be effective against dengue, resulting in the identification of 2964 small molecules, each assigned a unique ChEMBL identifier and SMILES code (Appendix A). Subsequently, 1627 unique compounds were selected from this dataset for virtual screening. Initially, the SMILES of these compounds were converted into 3D SDF files using the CACTUS tool [34]. These 3D SDF structures were converted into PDBQT files using the open Babel application [35]. Later, ensemble docking was performed with these 1627 selected anti-dengue compounds on cysteine proteinase protein.

### 2.4. Ensemble Docking

This study used ensemble docking to cover a larger conformational space on the selected structure of the MPXV cysteine proteinase produced after clustering. This ensemble docking used the central structure of the most populous clusters (members > 10% of the total population) from the MD simulation trajectory. These selected central structures of the protein were used to prepare the docking protocol. The hydrogen atoms were added to each atom’s protein structure and the charges using the AutoDock suite 4.2.6 and converted to the PDBQT file. The binding pockets of these selected structures were predicted in the CASTp web server [36]. The CASTp web server employs geometric and topological characteristics to anticipate the binding cavity of the protein structure. The CASTp can be used for its computational power to define active sites objectively, avoiding vagueness, and it depicts the binding pocket of the protein with high accuracy [37,38,39,40]. CASTp is used for the identification and measurement of all of the concavities in the protein; both pockets and voids. It identifies and measures both pockets and voids, and predicts the list of the residues with the SASA and volume of the pockets identified. Here, the best-ranked binding pocket was further used in the grid box design for docking. The docking was performed on each selected structure at its respective binding grid. The docking was performed using the AutoDock Vina software 1.2.0 [41]. Vina is a prominently cited and widely used docking software program licensed as free software for in silico research-based virtual screening. A study by Thomas Gaillard et al. asserts Vina as the main method for accurately reproducing experimental results [42]. The target protein, core cysteine proteinase, was subjected to molecular docking with 1627 anti-dengue compounds, using the selected binding site as input. During ensemble docking, a total of 20 binding modes were evaluated using default docking parameters, including an exhaustiveness value of 8 and a maximum energy difference of 4 Kcal/mol [43]. These parameters were employed to guide the docking protocols to determine the ligand’s most probable binding modes to the target protein. Following the docking study, the leading candidate compounds from each docking run were selected for additional analysis of their binding interactions using the PoseView component of the ProteinsPlus server [44]. Molecular dynamics simulations were then performed for each top hit in a complex with their respective docked structures, to gain further insight into their binding characteristics. Additionally, TTP-6171 was used as a control using the same protocol.

### 2.5. Molecular Dynamics (MD) Simulation of Protein–Ligand Complex

The top hits that emerged from ensemble docking outcomes underwent molecular dynamics (MD) simulations to explore the stability and flexibility of protein–ligand complexes [28,29]. The CHARMM36 force field was used to execute the force dynamics in the MD simulations of chosen complexes using the GROMACS2021 package. The CGenFF software version (interface is 1.0.0 and the force field is 3.0.1) generated topologies and compatible parameters with the CHARMM all-atom force field to prepare small molecules [30]. Electrostatic calculations were performed using the Ewald Particle Mesh [45]. After solvating the simulation box with the TIP3P water model and neutralizing it with Na+ and Cl- ions, the simulation was run. The protein–ligand solvated complex was positioned in the middle of the solvated box, 1 Å away from the wall. It was energetically minimized for 5000 steps using the steepest descent algorithm. All hydrogen bonds were constrained with the help of the LINCS algorithm, and the system was heated to 310 K [32]. The system was brought to a state of equilibrium under constant temperature (NVT) and pressure (NPT) conditions by maintaining a temperature of 310 K and a pressure of 1 bar, for a duration of 1 ns for each. The equilibrated system was used in the production run of 100 ns. The velocity-rescaling method was utilized for temperature coupling [46], while the Parrinello–Rahman pressure method was used to preserve the pressure [47]. The conformational analysis was carried out using various essential metrics provided in GROMACS.

### 2.6. MM/GBSA Calculations

The MM-GBSA (Molecular Mechanics Generalized Born Surface Area) technique was applied to calculate the binding free energy of the protein–ligand complex with the help of the gmx MMPBSA package [48,49]. The binding free energy ΔG for the top three hits was determined for the last 20 ns of the simulation trajectory. The salt concentration in the system was 0.154 M, while the solvation parameter (igb) was set to 5. Other parameters were used at their default values, which consist of the internal dielectric constants at 1.0 and the external dielectric constants at 80.0. These parameters were based on typical values used in several similar in silico studies [50,51].

Here, Equation (1) shows the MM-GBSA calculation method.
ΔG = <G_complex_ − [G_recepto_r + Gl_igand_]>(1)

Additionally, the < > sign represents the average binding free energy for the complex, the receptor, and the ligand over the last 20 ns simulation trajectory frames. Energetic components used in the ΔG calculation are determined using the Equations (2)–(6).
ΔG_binding_ = ΔH − TΔS(2)
ΔH = ΔG_GAS_ + ΔG_SOLV_(3)
ΔG_GAS_ = ΔE_EL_ + ΔE_VDWAALS_(4)
ΔG_SOLV_ = ΔE_GB_ + ΔE_SURF_(5)
ΔE_SURF_ = γ.SASA(6)

Moreover, ΔH is the enthalpy change composed of gas-phase energy (GGAS) and solvation-free energy (GSOLV). TΔS is the entropy contribution to the free binding energy. GGAS comprises electrostatic and van der Waals terms (EEL and EVDWAALS, respectively). GSOLV can be calculated from the polar solvation energy (GSOLV) and nonpolar solvation energy (ESURF), which can be further calculated from the product of solvent-accessible surface area (SASA) and solvent surface tension parameter (γ).

### 2.7. Machine Learning Models and Analogues Search

This study utilized the DeepPurpose architecture, a machine learning approach for predicting the interaction between ligands and target proteins [52]. The DeepPurpose pre-trained models were applied to predict drug–target interaction (DTI), and five pre-trained models were employed with distinct ligand protein encoding. These models were retrieved from the DeepPurpose repository on GitHub accessed on 10 March 2023 (https://github.com/kexinhuang12345/DeepPurpose). The model inputs were the amino acid sequence of the protein and the SMILES of the ligand compound, which were further encoded using different algorithms and fed into the prediction architecture. The IC_50_ prediction models from the DeepPurpose repository were selected, including CNN_CNN_BindingDB_IC_50_, Morgan_CNN_BindingDB_IC_50_, Morgan_AAC_BindingDB_IC_50_, MPNN_CNN_BindingDB_IC_50_, and Daylight_AAC_BindingDB_IC_50_. In these model names, the first encoder refers to the ligand encoding method, while the second refers to the protein-encoding technique. Subsequently, similar compounds were identified from the ChEMBL database [33] (https://www.ebi.ac.uk/chembl/ accessed on 5 July 2023) using a structure search criterion of 70% similarity, and their SMILES were collected. These SMILES were paired with protein sequences and used in the ML models to predict IC_50_ values.

## 3. Results and Discussion

### 3.1. Prediction of Protein Structure

The protein 3-dimensional structure of the MPXV core cysteine proteinase was predicted using the AlphaFold [25], as shown in Figure 1. It also shows the FASTA sequence (GenBank ID: UXF48136.1) used to predict the 3D structure. The fragment exhibiting the highest level of confidence as predicted by AlphaFold is visually represented in red, whereas the fragment with the lowest degree of confidence is depicted in blue. The gradient from red to blue represents the confidence level of prediction for each fragment in the 0% to 100% range. Figure 1 illustrates that most of the residues had a high confidence score. The overall prediction confidence score of the 3-dimensional structure of the MPXV core protein cysteine proteinase showed acceptable evidence to be used in this study. In this structure, a long loop is not appropriately modelled by AlphaFold, thus showing a low or medium confidence score. The model is prone to fluctuate in this region during and post-interaction with the ligand. However, the protein’s binding site was modelled accurately with a high confidence score. This provides high assurance for forming stable protein–ligand complexes in docking and molecular interaction during the molecular dynamic simulation. Overall, the protein structure is dominated by alpha helices. As shown in Figure 2a, the confidence score of the residues and their respective counts were plotted on the bar. It shows that most of the structural components of MPXV had high-quality scores given by Alpha Fold. Here, the confidence score ranging >70 indicates a high score, while <70 indicates a low score. Residues at the N- and C- terminals showed low confidence scores in prediction, which is widely accepted due to their high degree of freedom.

In addition to the confidence score given by AlphaFold, the structure quality assessment was performed using ProTSAV (Protein Structure Analysis and Validation) [26]. Figure 2b demonstrates the quality assessment of the modelled structure using ProTSAV, which utilizes various tools (such as Procheck, ProSA-web, ERRAT, Verify3D, dDFire, Naccess, MolProbity, D2N, ProQ, and PSN-QA) and metrics to evaluate protein structure quality. Except for PSN-QA, MolProbity, and ProSA-web, all the other tools showed high quality for the modelled protein structure. Notably, various quality assessment instruments employ distinct criteria and metrics to assess protein structure, and they may exhibit varying degrees of efficacy and limitations. PSN-QA uses a network-based approach to evaluate the quality of a protein’s 3D structure. The MolProbity tool analyses the overall geometry and stereochemistry of a protein structure.

In contrast, the ProSA web tool uses a potential statistical approach to predict the quality of a protein structurally. These tools may be more sensitive to certain types of discrepancy detected in the structure, while others may be less sensitive. This might be due to insufficient sampling during the modelling process. However, the other tools suggested that the quality of the protein structure was high (predicted RMSD < 2 Å), including Procheck, which calculates a set of stereochemical parameters, including the bond lengths, the bond angles, and the torsion angles, and compares them to traditional values to identify potential errors or complications within the structure, ERRAT quantifies the quality of a protein 3-dimensional structure by analyzing the agreement between the structure and the atomic environment around each residue. The high score in Procheck indicates the appropriate geometry of the model structure. However, it does not assure native confirmation. Verify3D uses a statistical approach to assess whether or not the structure of the protein is compatible with the sequence of its amino acids, dDFire uses a machine learning approach to investigate the quality of a protein structure based on structural features, including solvent accessibility, secondary structure, and packing density, whereas Naccess estimates the solvent accessibility of each residue in the protein structure, which can provide insight into the protein’s interactions with its environment.

Moreover, D2N (distance to native) predicts the modelled structure’s RMSD (root-mean-square deviation), which measures the structural similarity with the native contacts. ProQ uses machine learning and potential statistical approaches to predict a protein structure’s quality. The results indicated that most of these tools favored the quality of the protein structure, validating the modelled structure of MPXV core cysteine proteinase and, thus, progressing with further calculations.

A modelled protein structure of cysteine protease was used in structure-based drug design to screen phytochemicals against the MPXV [53]. This study discussed the application of molecular docking that identified N-(2-Allylcarbamoyl-4-chloro-phenyl)-3,4-dimethoxy-benzamide, which exhibited promising molecular binding results. A recent research study employed the structural model of cysteine protease as an approach to screening FDA-approved compounds [54].

### 3.2. MD (Molecular Dynamics) Simulation and Clustering of Protein Structure

Further, the protein structure of MPXV cysteine proteinase generated by AlphaFold was used in a molecular dynamics (MD) simulation, followed by clustering analysis to identify the most probable conformations. MD simulation has been widely used to refine the modeled protein structure [55]. Using water solvent molecules during the protein structure simulation has been recommended to refine the modelled conformation [56]. The criterion for selection stipulated that a cluster’s population must exceed 10% of the total conformations. Therefore, three clusters (Cluster-1, Cluster-2, and Cluster-3) were identified, accounting for most of the simulation trajectory. Cluster-1 had the highest number of conformations, with 65% of the total population; cluster-2 had a population of 16% of the total conformations; and cluster-3 had 11% of the members. These clusters collectively had more than 90% of the population, thus covering the major conformations of the protein structural change. This justified the selection of the top 3 representative structures through clustering. Each middle structure of these three clusters was selected for further calculation to cover the conformational variability of the binding site. They were docked individually using molecular docking to detect potential hit compounds. Here, the binding pocket was predicted using these three structures. Figure 3a,c,e shows the 3D representation of the middle structures of these selected clusters, Cluster-1, Cluster-2, and Cluster-3, respectively. Protein conformation largely changes with time, which can cause the movement of binding site residues. The multi-docking approach can address this challenge by considering different poses of the binding site [57]. However, enhanced sampling methods are recommended for more flexible proteins. This approach allowed us to cover the different states of the protein structure during docking.

### 3.3. Binding Pocket Prediction

The binding pockets of the selected middle structures (Cluster-1, Cluster-2, and Cluster-3) were predicted in the CASTp web server [36]. Here, the CASTp predicted different binding pockets for the three representative structures of MPXV cysteine proteinase generated during clustering (Cluster-1, Cluster-2, and Cluster-3), as shown in Figure 3. Table 1 lists the binding site residues of these binding pockets. The reason for the different binding pocket residues reflects the conformational changes in the protein structures; they are significant enough to cause changes in the geometrical characteristics of binding sites. It is also possible that the changes in conformation can affect the electrostatic properties and accessibility of the binding site, which can further contribute to differences in the predicted binding poses.

The binding pocket of Cluster-1 had an area of 515.961 Å^2^ and a volume of 417.746 Å^3^, Cluster-2 had an area of 537.845 Å^2^ and volume of 797.627 Å^3^ while Cluster-3 had an area of 411.089 Å^2^ and volume of 439.660 Å^3^. It was observed that Cluster-1 had the highest number of binding site residues compared to Cluster-2 and Cluster-3. However, Cluster-2 had the largest surface area and surface volume compared to Cluster-1 and Cluster-3. It was found that Tyr^4^, Tyr^276^ and Phe^278^ were common in all the binding site pockets of the clusters. This indicates that these pockets were distinct but not distant. Figure 3a,c,e showed the binding pocket of Cluster-1, Cluster-2, and Cluster-3, respectively, while Figure 3b,d,f showed the binding site residues of the Cluster-1, Cluster-2, and Cluster-3, respectively. A total of 39 residues were detected as the prime site in the cluster 1 binding pocket. However, out of the total, merely seven residues were found to be exclusively hydrophobic, while the majority of the residues were polar or charged in nature. Comparable patterns were noted in the remaining two clusters, albeit with a smaller number of residues. This indicates a high possibility of charged/polar ligands interacting with the target protein.

Generating ensembles of protein structures using MD simulation and detecting their binding sites accounts for the protein’s flexibility. This approach has been deployed earlier for drug discovery and resulted in a successful hit compound [58].

The grid box was built for each selected structure guided by its binding pocket (Cluster-1, Cluster-2, and Cluster-3) covering all their binding site residues. The binding sites were encompassed by grid boxes, where the dimensions of the grid box for Cluster-1 were 30 Å, 30 Å, and 40 Å in the x, y, and z axes, respectively, with the center of the box located at −8.102 Å, 8.790 Å, and −1.358 Å. Similarly, the dimensions of the grid box for Cluster-2 were 30 Å, 30 Å, and 32 Å in the x, y, and z axes, respectively, with the center of the box located at -1.823 Å, 10.733 Å, and 18.198 Å. Lastly, the grid box for Cluster-3 had dimensions of 30 Å, 30 Å, and 30 Å in the x, y, and z axes, respectively, with the center of the box located at 3.273 Å, 9.693 Å, and 17.796 Å, encompassing the binding sites. These grid boxes were used for docking with the anti-dengue compounds. As can be observed, the grid box for the Cluster-1 structure had the maximum dimension with an 8–10 Å edge in the Z-direction.

Previous studies stated that the cysteine proteinase enzyme consists of the putative catalytic triad residues, histidine (His) and cysteine (Cys), with the additional residue aspartate (Asp) embedded in a conserved region of the protein structure [59]. A catalytic triad is a group of three amino acids present at the active site of cysteine proteases that are essential for catalytic activity. The catalytic triad of MPXV core cysteine proteinases His^241^, Cys^328^, and Asp^248^ is shown in Figure 4, as are all three clusters. These catalytic residues are shared among the cysteine proteinases in this family. Moreover, these were detected in the binding grid box used for molecule docking in this study. Figure 4 shows the orientation and alignment differences among all of the cluster structures.

### 3.4. Ensemble Docking

In this study, the efficacy of anti-dengue compounds was evaluated against three distinct clusters of MPXV cysteine proteinase using the ensemble docking approach. The clusters were screened separately, and for each cluster, the middle structure was prepared and then docked with 1627 unique anti-dengue compounds. However, some compounds’ docking was unsuccessful due to conformational errors, resulting in only 1467 docking results. Appendix A lists the binding scores of the 1467 compounds and ranks them accordingly. The top hit was selected from docking for a given cluster, and their best pose was used for further interaction analysis. Table 2 lists the binding energies of the top hit from each cluster given by Autodock in kcal/mol. Cluster-3 docked with the compound CHEMBL4861364 had the best binding energy of −10.9 kcal/mol, compared to the other two with their respective clusters. Here, the compound CHEMBL4549312 had a minimum binding energy of −10.5 kcal/mol for Cluster-1, while CHEMBL32926 had a binding energy of −10.7 kcal/mol for Cluster-2. It was observed that the protein–ligand complex of Cluster-3 had the highest negative binding score, Cluster-2 was the second best, and Cluster-1 had the lowest negative binding score. All of the binding scores were <−10 kcal, showing a high affinity of these compounds with the target protein and, therefore, being good candidates for further development and investigation. TTP-6171 (control) was also docked with the same protocol with the three clusters as shown in Appendix A. Cluster-2 showed the highest negative binding score of −8.9 kcal/mol; thus, it was selected for further investigation. A similar study was performed with natural products, and the top-hit compound showed −10.56 kcal/mol of binding energy [60]. Based on these findings, it seems reasonable to recommend conducting additional research into whether these compounds could be useful in combating MPXV cysteine proteinase. The best hits from repurposing the FDA drugs showed binding scores between −9.0 and −7.0 kcal/mol [18]. Here, from a previous study, the known inhibitor TTP-6171 had a binding score of −9.3 kcal/mol [18]. This indicates that all three hits captured in this study have a higher binding potential.

### 3.5. Interactions Analysis

The top three hits’ best-docked poses of the protein–ligand complex were investigated for interaction analysis. The binding interactions were predicted using the PoseView component of the ProteinsPlus server [44]. Figure 5 shows the 2- and 3-dimensional representations of the binding interactions between the protein MPXV cysteine proteinase and the hit compounds. Here, it was observed that the compound CHEMBL4549312 had two hydrogen bonds with the residues Gly^261^ and Asp^362^ of Cluster-1 of MPXV cysteine proteinase. It also had hydrophobic contact with residue Leu^323^ of the protein, as shown in Figure 5a. This was also reflected by its relatively low binding score in Table 2. CHEMBL4549312 is a peculiar structure that has a lower degree of freedom. This prevents it from covering the larger sample space and forming more interactions.

Moreover, the compound CHEMBL32926 had multiple hydrophobic contacts with its corresponding binding pocket residues Tyr^4^, Thr^18^, Phe^278^, Val^320^_,_ and Ala^361^, and a single hydrogen bond interaction with residue Thr^5^ (Figure 5b). This shows that CHEMBL4549312 had more hydrogen bonds than CHEMBL32926, but the overall interaction was lesser with the pocket-lining residues. Figure 5c showed four hydrogen bonds between CHEMBL4861364 and Cluster-3 protein with residues Phe^278^, Lys^351^, Ser^349^, and Arg^417^. It also showed multiple hydrophobic contacts with residues Leu^30^, Phe^278^, Lys^348^, Lys^351^, and Lys^352^. Figure 5d–f showed that the compounds were bound to each cluster’s different binding locations of the MPXV protein. The compound CHEMBL4549312 was relatively bulkier as compared to the other two hits. The compound CHEMBL32926 bound to Cluster-2 was compact in nature compared to the other two compounds, while CHEMBL4861364 bound to Cluster-3 was of moderate size with a high degree of freedom. Here, it was observed that CHEMBL32926 and CHEMBL4861364 had greater interactions (polar and non-polar) with the MPXV cysteine proteinase than the large compound CHEMBL4549312.

Additionally, the control TTP-6171 was used for interaction analysis as shown in the Appendix A. Two hydrogen bonds were formed between the ligand and the protein with Thr^5^ and Arg^3^. Multiple hydrophobic contacts were formed with the residues Val^8^, Thr^18^, Phe^133^, Ala^361^ and Leu^360^.

### 3.6. ADMET Properties

In this study, we estimated the ADME-toxicity properties of the top three hits using the Swiss ADME server [61] and the ProTox-II (Prediction of Toxicity of Chemicals) server [62]. The ADME-toxicity properties of the hit compounds are summarized in Table 3. Here, it was observed that the molecular weight (MW) of CHEMBL32926 was less than 500 Da, while CHEMBL4549312 and CHEMBL4861364 had a greater molecular weight (>500 Da). The compound CHEMBL4549312 has a molecular weight of 1432.27 Da, indicating a considerable size exceeding the range typical of small molecule drugs. This indicates its lower flexibility inside the binding site of the protein. The hit compounds CHEMBL32926 and CHEMBL4861364 had less than 10 hydrogen bond acceptors and less than 5 hydrogen bond donors, whereas CHEMBL4549312 had a higher range of both acceptors and donors. These results suggest that CHEMBL32926 and CHEMBL4861364 have drug-like properties, particularly regarding their potential for oral bioavailability. The compounds CHEMBL4549312 and CHEMBL4861364 had seven rotatable bonds, indicating good flexibility. However, CHEMBL4549312 had a bulkier group attached, making moving within the binding site hard. All hit compounds had Ilogp values less than or close to 5, which is consistent with drug-like properties. The solubility of CHEMBL32926 was moderate, while that of CHEMBL4861364 was poorly soluble, and that of CHEMBL4549312 was insoluble. The observed complexity of CHEMBL4861364 suggests potential challenges in its ability to bond hydrogen with water molecules during the solvation process. The hit compounds had high gastrointestinal (GI) absorption for CHEMBL32926 but low GI absorption for CHEMBL4549312 and CHEMBL4861364. GI absorption aligns with solubility properties. The Lipinski rule of five was violated for CHEMBL4549312 and CHEMBL4861364 but not for CHEMBL32926, indicating the high drug-likeness of the latter. There was no PAINS alert for any hit compounds, indicating the hit compounds’ successful outcomes in in vitro and in vivo studies. According to the Hazard Communication Standard (HCS), which is based on the Globally Harmonized System of Classification and Labeling of Chemicals, it is permissible to classify all substances as belonging to the toxicity classes 4–5. The analysis of toxicity determined this. These findings suggest that the hit compounds CHEMBL32926 and CHEMBL4861364 have promising ADMET properties for further studies. However, both binding pattern and ADMET properties showed non-drug-like properties for CHEMBL4549312.

### 3.7. Molecular Dynamics Simulation

Molecular dynamics (MD) simulation was used in this investigation to ascertain the protein–ligand binding interaction strength and to draw conclusions. The root mean square deviation (RMSD) and root mean square fluctuation (RMSF) for the MD trajectory provide essential information on the system’s degree of adaptability in protein–ligand complexes. In the last stage of this process, the binding site residues were broken down into their parts and examined individually for how they interacted with the top three hit compounds.

#### 3.7.1. Root Mean Square Deviation (RMSD)

The highest-ranking ligands identified from molecular docking were evaluated for their binding stability to the MPXV core cysteine proteinase. The degree of conformational change induced by the protein–ligand interaction was measured using the root mean square deviation (RMSD) calculation. Appendix A shows the RMSD of the protein Cα-atoms and the ligand for the protein–ligand complex of Cluster-1 and CHEMBL4549312. The RMSD of the protein Cα-atoms was stable at the initial conformation until 0.15 ns and increased to 0.45 ns, where it fluctuated between the ranges of 0.4 nm to 0.5 nm. However, the RMSD of the ligand (calculated concerning protein) CHEMBL4549312 showed high fluctuations in the latter half of the 100 ns MD simulation. It was observed that, for the first 50 ns, the RMSD of the ligand was <0.5 nm; however, it increased with a jump to 1 nm and heavily fluctuated between 0.4 nm and 1 nm for the rest of the simulation. The analysis showed that the ligand CHEMBL4549312 exhibited an unstable binding towards the protein MPXV core cysteine proteinase and, hence, it was excluded from further experimental investigation. The structural conformation was impeded by its voluminous dimensions, thereby hindering the attainment of its optimal stability and leading to a limited duration of persistence. Figure 6a,c shows the root mean square deviation of the protein Cα atoms when the protein is in the bound state with the compounds CHEMBL32926 and CHEMBL4861364. After ligand binding, the simulation demonstrated cluster-2 and CHEMBL32926 complex protein structure changes. It was observed in Figure 6a that the RMSD of the protein Cα-atoms was 0.15 nm at the initial conformation, while it increased to 0.4 nm to 0.5 nm by the 60 ns simulation timeframe. However, it became stable for the last 30 ns of the simulation, with an RMSD of 0.4 nm. The ligand’s binding can change the protein’s conformation if the ligand binding site is located in a flexible protein region. Moreover, it is noted that the protein structure is the modelled structure and thus could be more prone to conformational changes than the experimental crystal structure. The RMSD of the protein Cα-atoms for the compound CHEMBL4861364 had an RMSD of 0.3 for the first 20 ns, increased to 1 nm at 40 ns, and fluctuated between 0.5 nm and 1 nm for the rest of the simulation, as shown in Figure 6c. This showed more fluctuation than the Cluster-2 complex, which indicates a higher effect of the binding of the ligand at the given binding site of the protein. Additionally, the RMSD of the protein and ligand was calculated for the control TTP-6171, as shown in the Appendix A. It was observed that the protein had stable RMSD for the majority of the simulation as shown in Appendix A. Initially, the RMSD of the protein was 0.2 nm which gradually increased to 0.4 nm and with dip decrease to 0.2 nm at 40 ns. After 40 ns, the RMSD of the protein was consistent with 0.4 nm for the rest of the simulation. The RMSD of the ligand showed high fluctuation for the first 60 ns simulation as shown in Appendix A. It increased from 0.5 nm to 1 nm and eventually to 1.5 nm till 60 ns; however, post 60 it decreased to 1 nm and stayed stable and consistent with no major fluctuations for the rest of the simulation. This indicated that the ligand TTP-6171 (control) changed its conformation; however, it had a stable binding for the last 20 ns simulation.

Later, the RMSD of the ligand showed that the compounds CHEMBL32926 and CHEMBL4861364 were stable compared to the compound CHEMBL4549312. Figure 6b showed a high increase in the RMSD at the 5 ns, 40 ns and 85 ns simulation with RMSD of >1 nm; however, the RMSD was consistent for maximum frames of the 100 ns simulation. This indicated that the compound CHEMBL32926 bound to the protein deviated from its initial docked conformation at the 5 ns, 40 ns and 85 ns simulation, as shown in Figure 6b. However, it stabilized in patches during the simulation. The RMSD of the ligand CHEMBL4861364 had a relatively high stability and consistency in RMSD for the 100 ns simulation, as shown in Figure 6d. It was observed that the RMSD was 0.3 nm to 0.4 nm for the first 25 ns, where it increased gradually to 0.6 nm, which then stabilized at 0.5 nm for the remainder of the simulation. However, the compound CHEMBL4861364 showed a high jump in RMSD at the 10 ns, 45 ns and 70 ns, where it deviated from its initial conformation, as shown in Figure 6d. Similarly, the control was analyzed at the 60 ns, 70 ns, 80 ns, and 100 ns to validate the change in conformation of the ligand to stabilize its binding with the protein.

#### 3.7.2. Interaction Analysis (MD)

The RMSD plots showed some deviations shown by the ligands CHEMBL32926 and CHEMBL32926 during the 100 ns simulation. The conformations at the 5 ns, 40 ns, and 85 ns timeframes of the simulation showed high deviation from the initial state for the compound CHEMBL32926 bound to the protein, while the 10 ns, 45 ns, and 70 ns timeframes also showed deviation for the compound CHEMBL4861364 bound to the protein. Thus, these conformations were extracted and compared to the initial states of the protein–ligand complex of Cluster-2 and CHEMBL32926 and Cluster-3 and CHEMBL4861364, respectively. Figure 7 and Figure 8 show the 3D conformations and the interactions of the complexes at the deviated state, along with the initial states of the complexes for CHEMBL32926 and CHEMBL32926, respectively.

The results presented in Figure 7 suggest that the binding interactions between the ligand CHEMBL32926 and Cluster-2 vary throughout the molecular dynamics (MD) simulation. Specifically, it was observed that the initial phase of the simulation had the highest number of bonds formed between the ligand and the protein, which gradually decreased at the 5 ns timeframe and then increased again at the 40 ns timeframe. This behavior is likely due to the changing conformations of both the protein and the ligand throughout the entirety of the simulation. Figure 7c shows that the initial state had two hydrogen bond interactions with residues Thr^5^ and Lys^358^. However, as the conformation of the ligand changed (Figure 7b), the ligand interactions cut down at the 5 ns timeframe, and there were no hydrogen bond interactions with either of these residues at 5 ns or 85 ns (as shown in Figure 7d,h). At 40 ns, only one hydrogen bond interaction was observed with residue Val320 (Figure 7g). These findings suggest that the binding interactions between the ligand and the protein are highly dependent on the conformational changes that occur in both the protein and the ligand throughout the MD simulation. The decrease in hydrogen bond interactions at 5 ns and 85 ns may indicate that the ligand has adopted less favorable conformations for binding to the protein. However, the increase in interactions at 40 ns suggests that the ligand may have adopted a more favorable conformation that is conducive to binding.

Further analysis may be necessary to determine the specific factors contributing to these observations. The findings indicate that, during the simulation, the complex of Cluster-3 and CHEMBL4861364 had a variable number of hydrogen bond interactions with the protein residues. At 0 ns, the complex formed four hydrogen bonds with residues Ser^349^, Thr^355^, Lys^352^, and Arg^417^. However, as the simulation progressed, the number of hydrogen bond interactions decreased, with only one hydrogen bond interaction observed at the 45 ns timeframe with residue Asp^421^. At the 70 ns timeframe, the compound showed two hydrogen bonds with the protein residues Glu^423^ and Ile^418^. The decrease in the number of hydrogen bond interactions was found to be correlated with changes in the conformation of the protein. Interestingly, the compound CHEMBL4861364 showed greater hydrogen bond interactions than the compound CHEMBL32926, which may explain why CHEMBL4861364 showed better performance overall, as observed in the RMSD fluctuation and molecular docking results. These findings suggest that the specific chemical structure of CHEMBL4861364 may be more favorable for binding to the protein and forming hydrogen bonds with key residues, resulting in a more stable and effective complex formation during the simulation.

Appendix A shows the conformations of the ligand TTP-6171 (control) bound to the protein at different timeframes during the MD simulation. This also includes the interactions formed between the protein and the ligand during the MD simulation. As shown in the Appendix A, the control at the 60 ns timeframe was bound at the cleavage region of the protein; however, over the course for the next 10 ns, the conformation of the ligand changed. At 70 ns, TTP-6171 shifted to a different region of the protein, still bound to it as shown in the Appendix A. Post-70 ns, the ligand moved to the inner cavity of the protein and became stable for the rest of the simulation as shown at the 80 ns, 90 ns, and 100 ns (Appendix A,i). Similarly, the interaction analysis showed that the ligand bound to the protein during the 60 ns and 70 ns had no interaction however, at the 80 ns and 90 ns timeframe, a single hydrogen bond was formed between the ligand and the protein, with the residue Glu14 indicating stable binding with the protein.

#### 3.7.3. Root Mean Square Fluctuation (RMSF)

The root mean square fluctuations, or RMSF values, were computed regarding the protein and the ligand molecules following their binding, to determine the individual fluctuations of each residue or atom in the bound state.

Figure 9 depicts the RMSF of the protein and ligand for complexes composed of Cluster-2 and CHEMBL32926, and Cluster-3 and CHEMBL4861364, respectively. Figure 9a,c displays the RMSF of the protein MPXV cysteine proteinase from the protein–ligand complexes of the selected compounds. Throughout the molecular dynamics (MD) simulation, the RMSF of proteins displayed comparable lower fluctuations, with RMSF values of less than 0.3 nm in both complexes. Nonetheless, a major peak was observed in the protein bound to CHEMBL32926, with an RMSF of 0.5 nm and 51 residues showing RMSF values of over 0.3 nm. Additionally, the protein bound to the compound CHEMBL4861364 exhibited greater RMSF fluctuations. Figure 9b,d shows the RMSF of the ligands, the chosen compounds CHEMBL32926 and CHEMBL4861364. The protein–ligand complex of CHEMBL32926 had no atoms with an RMSF of over 0.3 nm, while the compound CHEMBL4861364 had four residues with an RMSF of over 0.3 nm. Both compounds exhibited lower fluctuations, indicating the potential for stable binding to the MPXV protein. Appendix A showed the RMSF of the protein and ligand for TTP-6171 (control). It was observed that the protein had 40 residues showing RMSF values over 0.3 nm among them five residues had RMSF over 0.8 nm. The RMSF of the ligand was less than 0.4 nm during the MD simulation. Here, the RMSF of the control was comparable with the compounds with a high peak for the protein, as shown in Appendix A.

### 3.8. MM/GBSA Analysis

Protein–ligand complexes for compounds CHEMBL32926 and CHEMBL4861364 showed deviation at a certain timeframe of the simulations (as discussed in RMSD). These conformations were extracted, and their MMGBSA binding energies were determined to evaluate the binding of the extreme poses. The total free binding energy for the top two hits, which are listed in Table 4, was computed with the help of the tool known as the gmx MMPBSA test [49]. It was observed that the compound CHEMBL4861364 had a stronger total binding free energy than the compound CHEMBL32926. The initial docked complex of Cluster-2 with CHEMBL32926 and Cluster-3 with CHEMBL4861364 had positive total binding free energy. However, the other conformations at 5 ns, 40 ns, and 85 ns were 19.25 kcal/mol, 13.57 kcal/mol, and −9.65 kcal/mol, respectively, for CHEMBL32926. It was observed that the conformations at 10 ns, 45 ns, and 70 ns showed better total binding free energy for the complex Cluster-3 and CHEMBL4861364. The conformation at 45 ns had the best total binding free energy of −43.24 kcal/mol, while the complex at the 70 ns timeframe had −41.66 kcal/mol, and the 10 ns complex had −31.47 kcal/mol. This indicated that the compound CHEMBL4861364 had relatively better binding to the protein MPXV cysteine proteinase, compared to the compound CHEMBL32926. The magnitude of MMGBSA energy for CHEMBL4861364 was 3-fold compared to CHEMBL32926. This was demonstrated by more hydrogen bonds between the ligand and the protein during the simulation of the protein–ligand complex. These findings suggest that CHEMBL4861364 has a stronger binding affinity and inhibitory potential against MPXV cysteine proteinase and could be a promising candidate for experimental testing. Additionally, the control had total free binding energy of −32.30 kcal/mol indicating that the compound CHEMBL4861364 had better binding than the control, TTP-6171.

In a previous investigation, the natural ligand TTP-6171 was docked with the protein and its complex was simulated for 100 ns. This study reported ΔGTotal = −53.86 kcal/mol, which is very compatible with the binding free energy showed by CHEMBL486136453 [60].

### 3.9. IC_50_ Prediction and Analogue Search

Molecular docking and dynamic simulation techniques were employed to identify two potential hit compounds against a target protein. Subsequently, DeepPurpose, a machine learning-based activity prediction algorithm, was used to validate the binding scores of these compounds. Pretrained models were utilized in this analysis, and five distinct encoding method pairs were used for the protein and ligand. The corresponding encoding methods for these models are listed in Table 5. The models use the protein sequence and SMILES to predict their binding affinities. Six encoding techniques were employed: CNN, Morgan, AAC, MPNN, Daylight, and AAC. It is worth noting that Morgan and Daylight methods were used specifically for the ligand compound, while AAC was used exclusively for the protein sequence.

Five different models were employed to predict the binding affinity of the cysteine proteinase amino acid sequence and SMILES of CHEMBL32926 and CHEMBL4861364. These models utilized the IC_50_ concentration required to inhibit the protein’s function by 50% as an indicator of binding affinity, where a lower binding score (IC_50_) indicates a higher affinity. Table 6 shows the binding scores that each model predicted for the compounds CHEMBL32926 and CHEMBL4861364. Subsequently, all individual binding scores were aggregated, and an average score was calculated, which is referred to as the final binding score (shown in the last row of Table 6). Table 6 indicates that both compounds exhibit comparable predicted IC_50_ values, which strongly implies their potential binding affinity for the target protein due to the low IC_50_ observed.

Further, the close analogs for these compounds were searched in the ChemBL database with a 70% similarity threshold. These analogs are shown in Figure 10 and Figure 11 for compounds CHEMBL32926 and CHEMBL4861364. Two similar analogs for CHEMBL32926 were identified with 78.26% and 75.60% similarity, as shown in Figure 10.

Similarly, 70% of similar structures were searched for the CHEMBL4861364 compound, resulting in five analogs, as shown in Figure 11. These are 83.01%, 71.42%, 75.47%, 70.9%, and 70.17%, similar to the parent compound.

Later, these similar compounds were tested using the five DeepPurpose ML models to predict their IC_50_. Table 7 shows the average IC_50_ of the two analogues of CHEMBL32926 and the five analogues of CHEMBL4861364.

As shown in Table 7, CHEMBL1532491 showed a better IC_50_ of 5.22 nM than its parent compound (CHEMBL32926), while CHEMBL4850198 was the best analogue detected for CHEMBL4861364, with an IC_50_ of 5.84 nM. Moreover, the other two analogs, CHEMBL82742 and CHEMBL4868418, also showed high binding with IC_50_s of 5.97 nM and 5.99 nM for the target protein, as detected by the ML models. Analog findings added four more compounds with similar structures and binding to the target protein.

## 4. Conclusions

The present investigation explores anti-dengue compounds’ binding affinity with the monkeypox virus’s cysteine proteinase. The study employs various in silico methods and techniques to screen and evaluate the performance of the screened compounds. An ensemble effect was created by using multiple conformations of the protein structure and its corresponding binding pockets. This effect was then put through additional testing with the help of molecular dynamic simulations. The goal of this was to increase the accuracy of the results. This approach accounted for the flexibility of the protein structure and its ability to adopt different conformations, which can affect the binding site and affinity of potential inhibitors. Two potential MPXV core cysteine proteinase binders were identified, with ChemBL IDs CHEMBL32926 and CHEMBL4861364 showing strong binding scores, with multiple hydrogen bonds during docking. These compounds also demonstrated drug-like properties, calculated using ADMET tools, making them potential candidates for experimental testing. During the simulation, their bindings were retained, further indicating their stable complex formation with the protein. The consistent proximity of these compounds with the catalytic dyad of the protein implies their inhibitory action upon binding. The later half of the study was focused on identifying the analogues of the detected hit compounds and their IC_50_ predictions. This showed that two compounds, each for CHEMBL32926 and CHEMBL4861364, were identified as promising analogues that can equally inhibit the function of cysteine proteinase in the monkeypox virus.

## Figures and Tables

**Figure 1 biomedicines-11-02025-f001:**
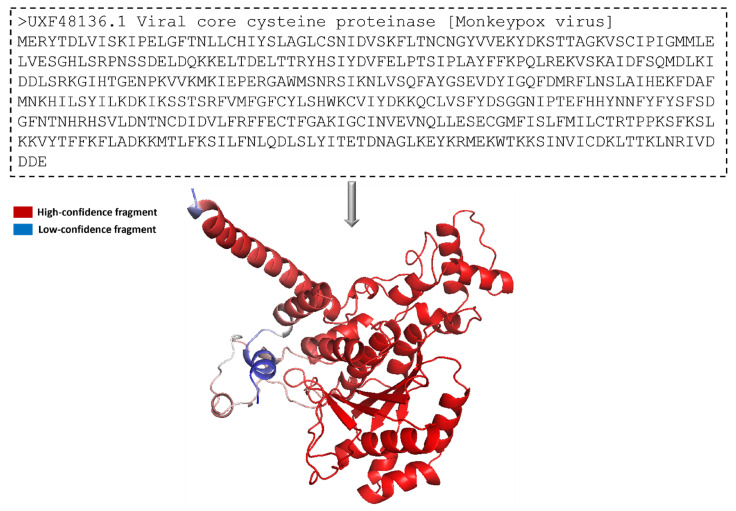
Three-dimensional representation of the predicted modeled structure of cysteine proteinase from MPXV, predicted using the AlphaFold program.

**Figure 2 biomedicines-11-02025-f002:**
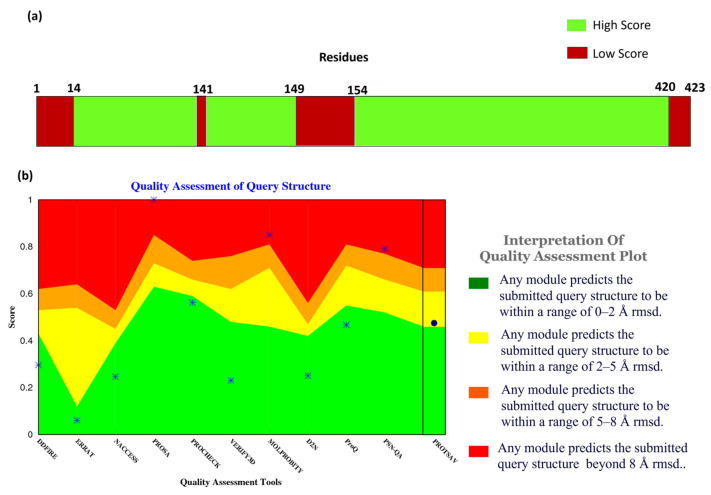
Quality assessment of the modeled structure: (**a**) Confidence scores of the residues of the modeled 3D structure of cysteine proteinase and (**b**) Quality assessment plot of the modeled 3D structure of cysteine proteinase. (RMSD is given in Å).

**Figure 3 biomedicines-11-02025-f003:**
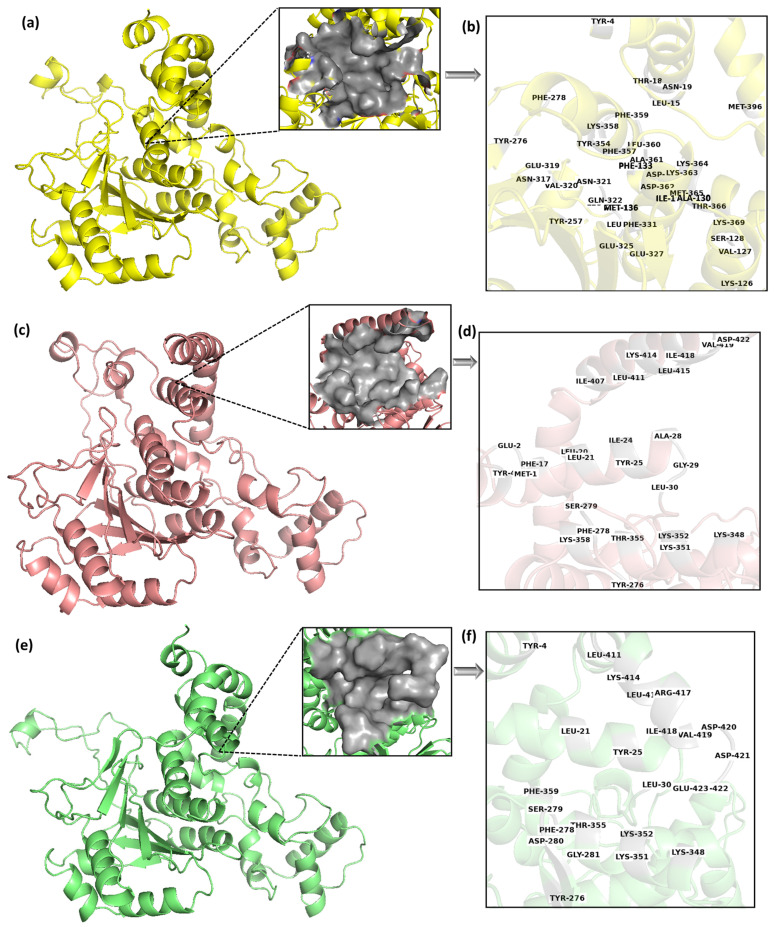
The middle structure of the three selected clusters with their corresponding binding pockets and binding site residues, (**a**,**b**) Cluster-1, (**c**,**d**) Cluster-2 and (**e**,**f**) Cluster-3.

**Figure 4 biomedicines-11-02025-f004:**
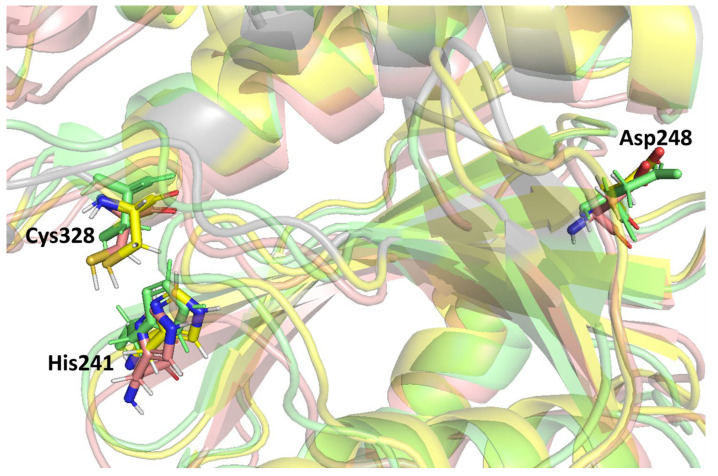
The catalytic triad residues of MPXV core cysteine proteinase in the three structures selected from clustering (Cluster-1, Cluster-2, Cluster-3).

**Figure 5 biomedicines-11-02025-f005:**
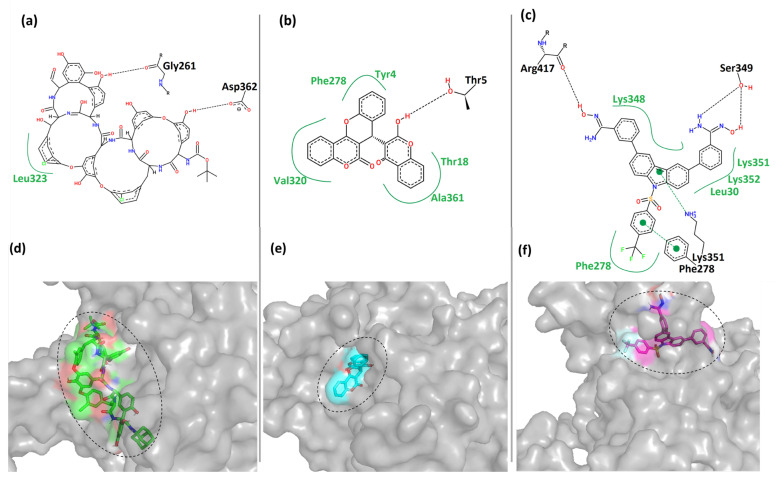
The 2D and 3D representations of the binding interactions of the complexes (**a**,**d**) Cluster-1 and CHEMBL4549312 (**b**,**e**) Cluster-2 and CHEMBL32926 (**c**,**f**) Cluster-3 and CHEMBL4861364. Compounds are illustrated in different colors.

**Figure 6 biomedicines-11-02025-f006:**
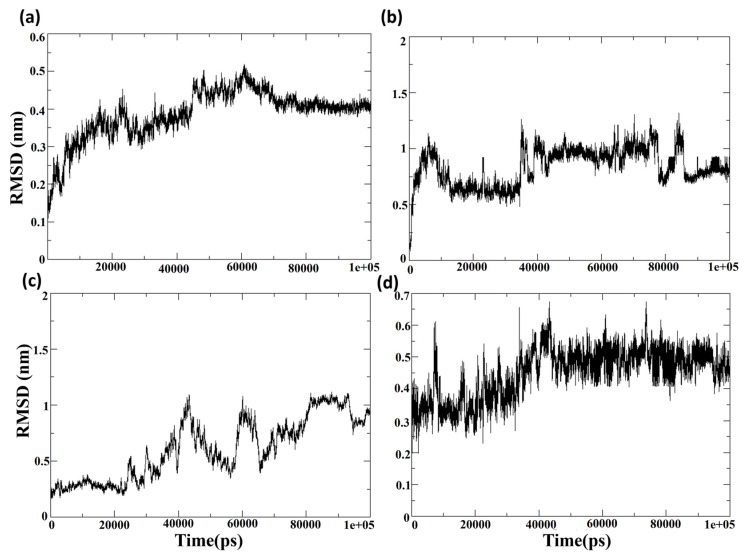
The RMSD of the protein Cα-atoms and the ligand for the protein–ligand complex of (**a**,**b**) Cluster-2 and CHEMBL32926 and (**c**,**d**) Cluster-3 and CHEMBL4861364.

**Figure 7 biomedicines-11-02025-f007:**
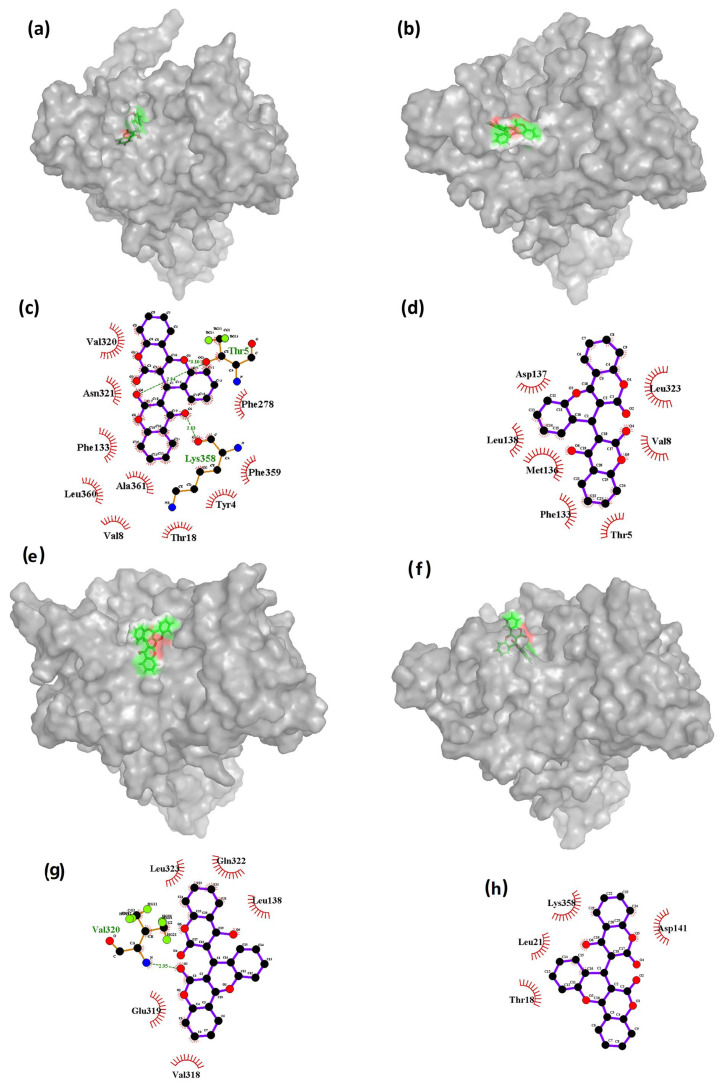
The 3D and 2D representations of the binding interactions of the Cluster-2 and CHEMBL32926 complex during the 100 ns MD simulation at the (**a**,**c**) 0 ns, (**b**,**d**) 5 ns, (**e**,**g**) 40 ns, and (**f**,**h**) 85 ns conformations.

**Figure 8 biomedicines-11-02025-f008:**
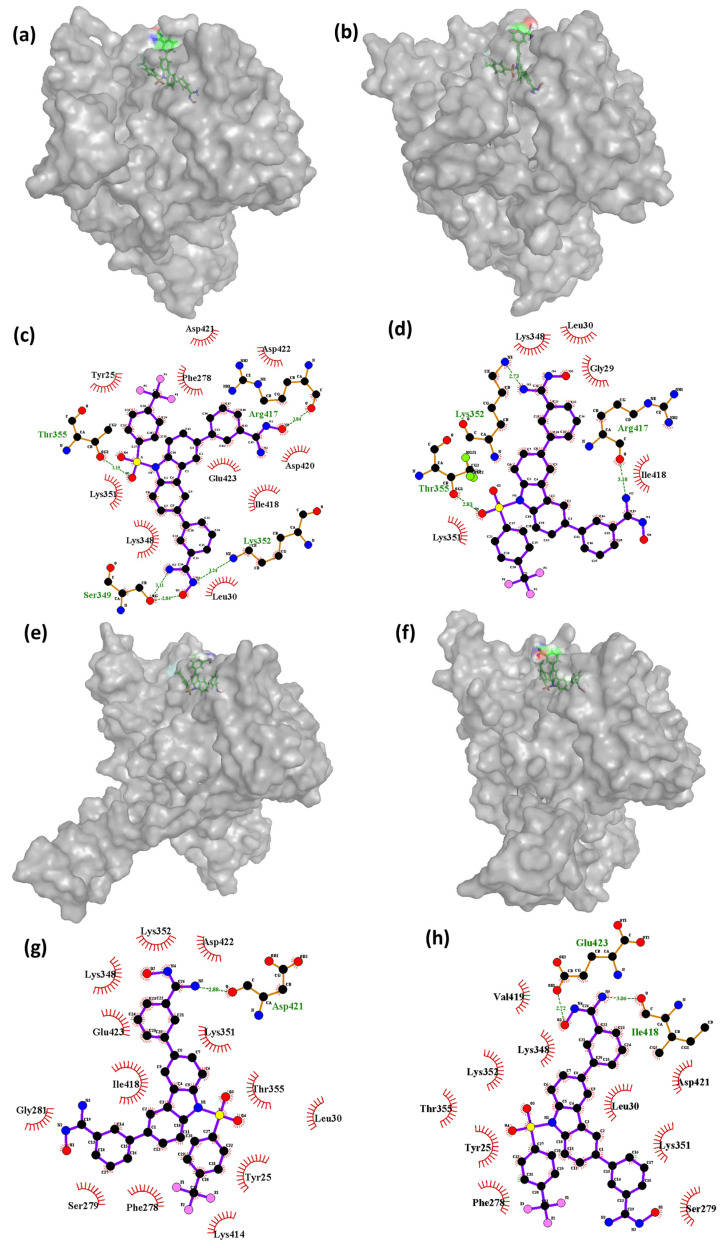
The 3D and 2D representations of the binding interactions of the Cluster-3 and CHEMBL4861364 complex during the 100 ns MD simulation at the (**a**,**c**) 0 ns, (**b**,**d**) 10 ns, (**e**,**g**) 45 ns, and (**f**,**h**) 70 ns conformations.

**Figure 9 biomedicines-11-02025-f009:**
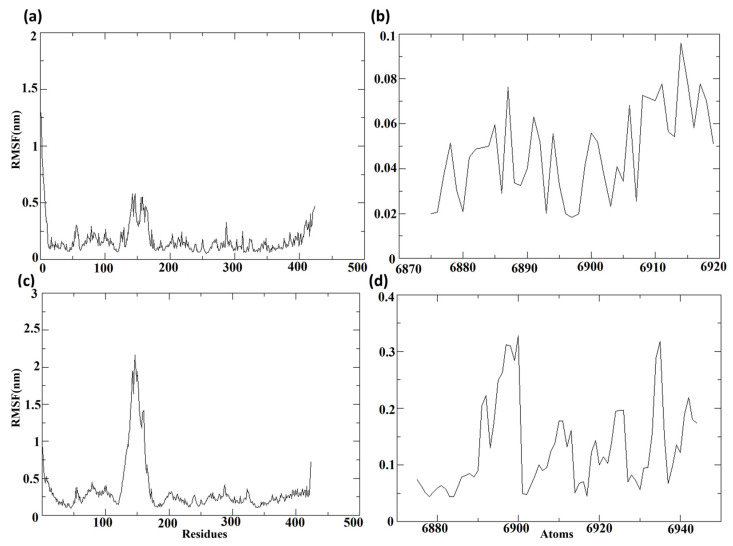
The RMSF of the protein and the ligand for the protein–ligand complex of (**a**,**b**) Cluster-2 and CHEMBL32926 and (**c**,**d**) Cluster-3 and CHEMBL4861364.

**Figure 10 biomedicines-11-02025-f010:**
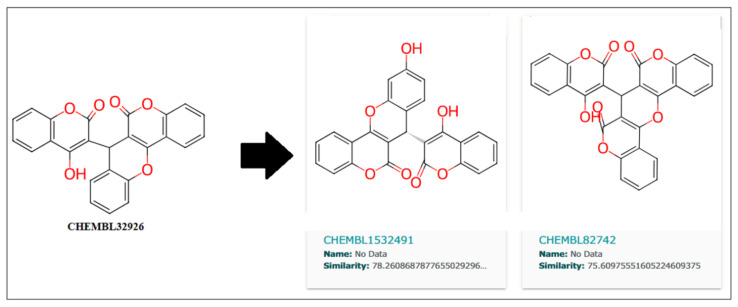
Two similar compounds of CHEMBL32926 with a similarity threshold of 70% were searched in the ChemBL database.

**Figure 11 biomedicines-11-02025-f011:**
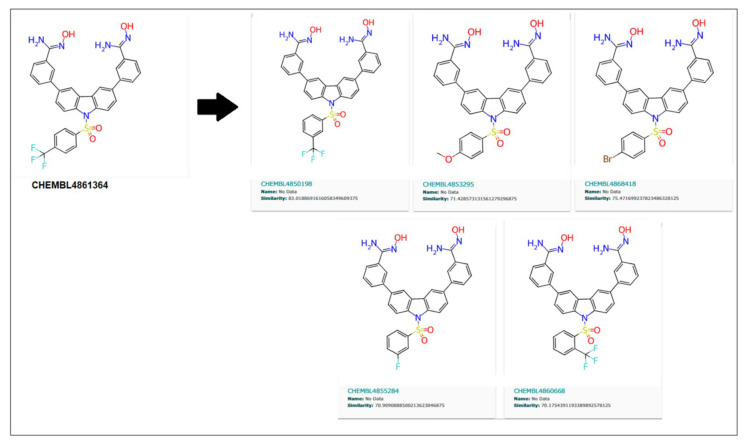
Five similar compounds of CHEMBL4861364 with a similarity threshold of 70% searched in the ChemBL database.

**Table 1 biomedicines-11-02025-t001:** Surface area, surface volume and binding site residues of the three clusters (Cluster-1, Cluster-2, Cluster-3) predicted from the CASTp server.

Clusters	Surface Area (SA) Å^2^	Surface Volume (SA) Å^3^	Binding Site Residues Predicted from the CASTp Server
Cluster-1	515.96	417.74	Tyr^4^, Glu^14^, Leu^15^, Thr^18^, Asn^19^, Lys^126^, Val^127^, Ser^128^, Ala^130^, Ile^131^, Asp^132^, Phe^133^, Met^136^, Tyr^257^, Tyr^276^, Phe^278^, Asn^317^, Glu^319^, Val^320^, Asn^321^, Gln^322^, Leu^323^, Leu^324^, Glu^325^, Glu^327^, Phe^331^, Tyr^354^, Phe^357^, Lys^358^, Phe^359^, Leu^360^, Ala^361^, Asp^362^, Lys^363^, Lys^364^, Met^365^, Thr^366^, Lys^369^, Met^396^
Cluster-2	537.84	797.62	Phe^17^, Leu^20^, Leu^21^, Ile^24^, Tyr^25^, Ala^28^, Gly^29^, Leu^30^, Tyr^276^, Phe^278^, Ser^279^, Lys^348^, Lys^351^, Lys^352^, Thr^355^, Lys^358^, Ile^407^, Leu^411^, Lys^414^, Leu^415^, Ile^418^, Val^419^, Asp^422^
Cluster-3	411.08	439.66	Leu^30^, Tyr^276^, Phe^278^, Ser^279^, Asp^280^, Gly^281^, Lys^348^, Lys^351^, Lys^352^, Thr^355^, Phe^359^, Leu^411^, Lys^414^, Leu^415^, Arg^417^, Ile^418^, Val^419^, Asp^420^, Asp^421^, Asp^422^, Glu^423^

**Table 2 biomedicines-11-02025-t002:** Autodock binding scores of the best hit compounds from docking of each cluster of the MPXV cysteine proteinase.

Clusters	Hit Compounds	Binding Energy (kcal/mol)
Cluster-1	CHEMBL4549312	−10.5
Cluster-2	CHEMBL32926	−10.7
Cluster-3	CHEMBL4861364	−10.9

**Table 3 biomedicines-11-02025-t003:** The ADME-toxicity study of the top hit compounds from each docking with MPXV cysteine proteinase.

ADMET Properties	CHEMBL4549312	CHEMBL32926	CHEMBL4861364
MW	1432.27	410.38	643.63
Rotatable bonds	7	1	7
H-bond acceptors	19	6	9
H-bond donors	15	1	4
MR	387.79	114.56	168.56
TPSA	411.33	89.88	164.67
Ilogp	5.74	3.33	3.2
Solubility Class	Insoluble	Moderately soluble	Poorly soluble
GI absorption	Low	High	Low
Lipinski violations	3	0	2
PAINS alerts	0	0	0
Predicted Toxicity Class	5	5	4
Predicted LD50	5000 mg/kg	2500 mg/kg	957 mg/kg

**Table 4 biomedicines-11-02025-t004:** The total binding free energy of the two selected hits at the conformations of their high RMSD peaks and their initial conformation from the 100 ns MD simulation.

CHEMBL32926	ΔG_TOTAL_ (kcal/mol)	CHEMBL4861364	ΔG_TOTAL_ (kcal/mol)
0 ns	44.18	0 ns	7.77
5 ns	−19.25	10 ns	−31.47
40 ns	−13.57	45 ns	−43.24
85 ns	−9.65	70 ns	−41.66

**Table 5 biomedicines-11-02025-t005:** Pretrained ML models sourced from DeepPurpose GitHub with their corresponding encoding method for ligand and protein.

Model Name	Ligand Encoding	Protein Encoding
Pred-I	CNN (Convolutional Neural Network on SMILES)	CNN (Convolutional Neural Network on Protein Sequence)
Pred-II	Morgan (Extended-Connectivity Fingerprints)	CNN (Convolutional Neural Network on Protein Sequence)
Pred-III	Morgan (Extended-Connectivity Fingerprints)	AAC (Amino acid composition up to 3-mers)
Pred-IV	MPNN (Message-passing neural network)	CNN (Convolutional Neural Network on Protein Sequence)
Pred-V	Daylight (Daylight-type fingerprints)	AAC (Amino acid composition up to 3-mers)

**Table 6 biomedicines-11-02025-t006:** Binding score (IC_50_) predicted by the five pre-trained ML modes for the top two hit compounds.

Models	Binding Score (IC50)	
	CHEMBL32926	CHEMBL4861364
Pred-I	5.74	7.23
Pred-II	4.8	4.76
Pred-III	4.88	4.78
Pred-IV	4.98	4.14
Pred-V	5.76	6.52
Final Binding Score (Average)	5.23	5.48

**Table 7 biomedicines-11-02025-t007:** Predicted IC_50_ of similar compounds of the two hits, predicted IC_50_ score is average of five scores calculated by five DeepPurpose pre-trained models.

CHEMBL ID	SMILES	Predicted IC_50_
Analogues (CHEMBL32926)		
CHEMBL1532491	O=c1oc2ccccc2c(O)c1[C@H]1c2ccc(O)cc2Oc2c1c(=O)oc1ccccc21	5.22
CHEMBL82742	O=c1oc2ccccc2c(O)c1C1c2c(c3ccccc3oc2=O)Oc2c1c(=O)oc1ccccc21	5.97
Analogues (CHEMBL4861364)		
CHEMBL4850198	N/C(=N\O)c1cccc(-c2ccc3c(c2)c2cc(-c4cccc(/C(N)=N/O)c4)ccc2n3S(=O)(=O)c2cccc(C(F)(F)F)c2)c1	5.84
CHEMBL4868418	N/C(=N\O)c1cccc(-c2ccc3c(c2)c2cc(-c4cccc(/C(N)=N/O)c4)ccc2n3S(=O)(=O)c2ccc(Br)cc2)c1	5.99
CHEMBL4853295	COc1ccc(S(=O)(=O)n2c3ccc(-c4cccc(/C(N)=N/O)c4)cc3c3cc(-c4cccc(/C(N)=N/O)c4)ccc32)cc1	7.25
CHEMBL4855284	N/C(=N\O)c1cccc(-c2ccc3c(c2)c2cc(-c4cccc(/C(N)=N/O)c4)ccc2n3S(=O)(=O)c2cccc(F)c2)c1	6.25
CHEMBL4860668	N/C(=N\O)c1cccc(-c2ccc3c(c2)c2cc(-c4cccc(/C(N)=N/O)c4)ccc2n3S(=O)(=O)c2ccccc2C(F)(F)F)c1	6.8

## Data Availability

Data is contained within the article or Appendix A.

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
