# Peer review of "Repurposing Anti-Dengue Compounds against Monkeypox Virus Targeting Core Cysteine Protease"

_biomedicines, 2023, doi:10.3390/biomedicines11072025_

Round 1

Reviewer 1 Report

It was a pleasure to review this article of importance in its field. I found the research question was clearly stated and well expressed objectives. The methods applied were seems to be appropriate and fully described in the article. Similarly, the results section well furnished, illustrative and self-explanatory enough, to answer the questions posed at the beginning. However, some minor important issues arise from the manuscript at its present state:

In the study, the readability of molecules should be increased, whether they interact with proteins or the shapes of molecular structures.

Reviewer 2 Report

The manuscript addresses the exploration of anti-dengue compounds' binding affinity with the monkeypox virus's cysteine proteinase and confirmed their activity by comparing it with compound library ChEMBL database.

This article has provided structure-based drug design approach to identify potential inhibitors for the core cysteine proteinase of MPXV along with their simulation and docking study. The authors have employed protein structure modelling, molecular dynamics simulation of modelled structure, ensemble docking, molecular dynamics (MD) simulation of protein-ligand complex, MM/GBSA calculations to elucidate activity of selected compounds.

General Remarks:

I would like to ask the authors about the rationale behind this study? On what basis did the authors select these compounds? Also, provide a supplementary file containing the structures of the screened compounds? Moreover, relevant literature should be cited to justify the selection of the compounds based on SAR from literature?

Other Minor Remarks:

Line 77 and 78, word proteinase has been repeated. Recheck.

Line 147, add space after word ‘’members’’.

Line 162, recheck word ‘’exhaustiveness’’

In heading ‘’Machine Learning Models and Analogues Search’’, use correct notation of IC50.

Line 235 & 236 needs to be re-written.

Line 250, add  space before ‘’70’’.

Line 251, word terminal is repeated.

Line 298 & 299, needs to be re-written.

Line 306 and 307 needs to be re-written.

Line 307, add space before ‘’10’’.

Line 345 & 346, rephrase the sentence.

In 2nd paragraph of ‘’ Binding Pocket Prediction’’, use proper notation for surface area and volume.

Line 363, remove extra space.

Line 422, remove extra space.

Line 427, add ‘’ as ‘’ before the word ‘’compared’’.

Line 443, add space before ‘’500’’.

Line 451, remove extra space.

Line 459, remove extra space.

Line 463-465, rephrase the sentence.

Line 473, root means square??

Line 605 & 611, remove extra space.

Line 624, rephrase the sentence.

Use correct notation of ‘’IC50’’  throughout the manuscript. 

The manuscript suffers from typos and grammatical errors. Authors need to thoroughly check the the English language and rectify them.

Reviewer 3 Report

In this study authors used structural-based drug design approach to identify potential inhibitors for the core cysteine proteinase of monkeypox virus. They used ensemble-based protein-ligand docking to account for the binding site conformation variability and molecular dynamic simulations of protein-ligand complexes showed fluctuation from the initial docked pose, but it confirmed their binding throughout the simulation.

The paper is well organized. Abstract is clear, introduction gave us a lot of information. Each step of methodology is explained in detail. Results are very well described.

No comments for revision.

Round 2

Reviewer 2 Report

The manuscript has now been revised and I am satisfied with the revision. The manuscript can now be accepted in current form.

Author Response

please refer the attached file
